# Wake-Up Free Ultrathin Ferroelectric Hf_0.5_Zr_0.5_O_2_ Films

**DOI:** 10.3390/nano13212825

**Published:** 2023-10-25

**Authors:** Anastasia Chouprik, Vitalii Mikheev, Evgeny Korostylev, Maxim Kozodaev, Sergey Zarubin, Denis Vinnik, Svetlana Gudkova, Dmitrii Negrov

**Affiliations:** 1Moscow Institute of Physics and Technology (National Research University), Institutskii per. 9, 141701 Dolgoprudny, Russia; mikheev.vv@phystech.edu (V.M.); korostylev.ev@phystech.edu (E.K.); kozodaev.mg@mipt.ru (M.K.); zarubin.ss@mipt.ru (S.Z.); vinnikda@susu.ru (D.V.); svetlanagudkova@yandex.ru (S.G.); dmitriynegrov@gmail.com (D.N.); 2Material Science and Physics & Chemistry of Materials, South Ural State University (National Research University), Lenin’s Prospect 76, 454080 Chelyabinsk, Russia; 3Institute of Chemistry, Saint-Petersburg State University, Universitetskaya Embankment 7-9, 199034 St. Petersburg, Russia

**Keywords:** ultrathin films, ferroelectric memory, ferroelectric tunnel junction, ferroelectric hafnium oxide, wake-up, piezoresponse force microscopy, synchrotron X-ray micro-diffraction

## Abstract

The development of the new generation of non-volatile high-density ferroelectric memory requires the utilization of ultrathin ferroelectric films. The most promising candidates are polycrystalline-doped HfO_2_ films because of their perfect compatibility with silicon technology and excellent ferroelectric properties. However, the remanent polarization of HfO_2_ films is known to degrade when their thickness is reduced to a few nanometers. One of the reasons for this phenomenon is the wake-up effect, which is more pronounced in the thinner the film. For the ultrathin HfO_2_ films, it can be so long-lasting that degradation occurs even before the wake-up procedure is completed. In this work, an approach to suppress the wake-up in ultrathin Hf_0.5_Zr_0.5_O_2_ films is elucidated. By engineering internal built-in fields in an as-prepared structure, a stable ferroelectricity without a wake-up effect is induced in 4.5 nm thick Hf_0.5_Zr_0.5_O_2_ film. By analysis of the functional characteristics of ferroelectric structures with a different pattern of internal built-in fields and their comparison with the results of in situ piezoresponse force microscopy and synchrotron X-ray micro-diffraction, the important role of built-in fields in ferroelectricity of ultrathin Hf_0.5_Zr_0.5_O_2_ films as well as the origin of stable ferroelectric properties is revealed.

## 1. Introduction

Due to the perfect compatibility with modern silicon technology, ferroelectric HfO_2_ polycrystalline thin films are the most promising functional material for the implementation of the next-generation non-volatile ferroelectric memory. One of the important trends in ferroelectric memory development relies on the utilization of ultrathin (of a few nm) ferroelectric films, which are desired for high-density ferroelectric random-access memory (FRAM) and non-volatile resistive memory based on ferroelectric tunneling junctions (FTJs). For inducing ferroelectric properties, the optimum thickness of hafnium oxide film is about 10 nm [1,2,3], and the remanent polarization degrades with decreasing film thickness below 10 nm. Therefore, the development of approaches to maintain the large remanent polarization during film scaling is of vital importance for both materials science and practical applications.

The critical thickness of ferroelectric hafnium oxide films depends on many causes. In the as-deposited state, HfO_2_ film is amorphous, and a subsequent annealing process is required to crystallize it. Hafnium oxide has several polymorphs, and the structural phase of its equilibrium is the paraelectric monoclinic *P*2_1_/c phase, whereas the ferroelectric orthorhombic *Pca*2_1_ phase is metastable. After crystallization, the HfO_2_ film becomes polycrystalline, and the fraction of ferroelectric grains depends on several parameters including annealing temperature, doping element and its concentration, oxygen vacancy density, and substrate/electrode materials [2]. When the film thickness decreases below 10 nm, the surface energy effect becomes dominant. Due to the increase in the ratio of grain surface energy to bulk energy, the formation of ferroelectric grain becomes energetically unfavorable. To date, several approaches have been proposed to compensate for this effect. In particular, it was shown that ferroelectricity in ultrathin films is promoted by increasing the annealing temperature budget. In addition, alloyed Hf_0.5_Zr_0.5_O_2_ (HZO) films are the best option because the minimal stabilization temperature for the ferroelectric phase is significantly lower than the annealing temperature for other doped HfO_2_ films (400 °C vs. 800 °C) [1,2,3]. Next, material engineering revealed that ferroelectricity in ultrathin films is promoted by using substrates and electrode materials with a low thermal coefficient of expansion. In this case, during annealing, the hafnium oxide film experiences significant mechanical stress that lowers the phase transition temperature from the high-temperature parent nonpolar cubic *P*m3m and tetragonal *P*4_2_/nmc phases to the polar orthorhombic phase [2,4,5,6]. Despite the progress in inducing ferroelectricity in ultrathin films, one important challenge has not yet been addressed—the prolonged wake-up effect.

The wake-up effect consists of the gradual increase and stabilization of the remanent polarization during the initial stage of the HfO_2_-based memory cell operation [7,8,9,10]. In ferroelectric HfO_2_, this effect mainly originates from the gradual reduction of the internal built-in fields existing in an as-prepared memory cell and the de-pinning of ferroelectric domains. Due to the reduction in the built-in fields, pinning gradually disappears, and more and more domains switch and contribute to the measured remanent polarization. There are several hypotheses to explain the nature of the built-in fields in ferroelectrics [11,12,13,14]. In particular, it has been shown that hafnium oxide easily enters oxidation–reduction (electro-) chemical reaction with electrode materials, accompanied by the formation of oxygen vacancies nearby the electrode interfaces [15,16]. These charges are the most likely source of the built-in fields in this material. The disappearance of the internal built-in fields is explained by the gradual redistribution of charged defects over the film bulk in response to the cyclic bipolar pulse application.

In ultrathin films, the wake-up effect severely hinders practical applications. Indeed, since the electric field strength is inversely proportional to the thickness of the ferroelectric layer, the built-in fields are larger in ultrathin films, and a much higher number of bipolar switching cycles are needed to eliminate them (approximately 10^5^ bipolar cycles for 5 nm thick HZO film and 10^3^–10^4^ cycles for 10 nm thick one [6,7,8,9,10]). On the other hand, naturally, ultrathin films have a lower endurance (10^5^–10^6^ bipolar switching cycles), i.e., the fatigue or even electrical breakdown of the memory cells occurs before the end of the wake-up procedure. Therefore, for the development of high-density FRAM and FTJ memory cells, it is necessary to elucidate the physical basis behind needed material and interface engineering and to develop the optimal technology for the fabrication of the wake-up-free ultrathin ferroelectric HZO films. In this work, we propose an approach to suppress the internal built-in fields in as-prepared memory cells based on ultrathin HZO films and induce stable ferroelectricity in a 4.5 nm thick HZO film. By analyzing the functional characteristics and comparing them with the results of in situ piezoresponse force microscopy (PFM) and synchrotron X-ray microdiffractometry (micro-XRD), we reveal the origin of the improved ferroelectric properties. The proposed approach could be useful for the material and interface engineering for the development of the next-generation low-power and high-density FRAM and FTJ memories.

## 2. Materials and Methods

*Structures fabrication*. Hf_0.5_Zr_0.5_O_2_ (HZO) films 4.0, 4.5, 5.0, and 10 nm in thickness are grown on preliminarily electroded silicon (100) substrates via thermal atomic layer deposition at a 280 °C substrate temperature using Hf[N(CH_3_)(C_2_H_5_)]_4_ (TEMAH), Zr[N(CH_3_)(C_2_H_5_)]_4_ (TEMAZ) and H_2_O as precursors and N_2_ as a carrier and a purging gas. The bottom electrode is either a 25 nm thick W film or a 25 nm thick TiN film, both deposited by magnetron sputtering. The top electrode consists of two layers. First, a 20 nm thick top TiN film is deposited via magnetron sputtering. Second, an additional Al layer, 150 nm in thickness, is deposited on the top of the TiN electrode using electron beam evaporation. Al layer minimizes the *RC* time constant and thus allows to decrease the duration of switching pulses, which, in turn, improves the endurance of ferroelectric devices. After all, patterning of the top electrode is subsequently performed using maskless optical lithography and plasma chemical etching.

Crystallization of HZO film is induced by either post-metallization rapid thermal annealing or post-deposition rapid thermal annealing at 500, 600, and 700 °C for 30 s in Ar atmosphere. During post-metallization annealing (PMA), HZO film is capped by the top TiN layer, whereas during post-deposition annealing (PDA), it is non-capped.

For the in situ PFM and micro-XRD studies, the functional capacitors were routed to the Al contact pads, allowing external electric biasing of the capacitors. The routing fabrication details were described previously [10].

For *ferroelectric characterization*, the Cascade probe station coupled with the semiconductor parameter analyzer B1500A (Agilent Technologies, Santa Clara, CA, USA) is used. Switching *I–V* and *P–V* curves are measured through the dynamic positive-up negative-down (PUND)-like technique as described below. To wake up the as-prepared HZO film, the ferroelectric capacitors are cycled 10^5^ times by applying bipolar voltage double triangular pulses with a duration of 10 µs and amplitude of ±4 MV/cm for 4.0–5.0 thick HZO films and ±3 MV/cm for 10 thick HZO films.

*Piezoresponse force microscopy.* Local piezoresponse is characterized via the in-house implemented resonance-enhanced band excitation PFM technique using an Ntegra atomic force microscope (NT-MDT Spectrum Instruments, Moscow, Russia) and a M3300A arbitrary waveform generator/digitizer (Keysight, Santa Rosa, CA, USA). The experimental scheme and the details of the BE PFM were described previously [10]. To minimize the contribution of the parasitic electrostatic tip–surface interactions, the PFM experiments are carried out at the patterned capacitors routed to the contact pads.

The electrical excitation of the ferroelectric layer is performed with the following waveform parameters: the central frequency near the contact resonance frequency was ~350 kHz, the bandwidth was 50 kHz with 256 frequency bins, and the peak-to-peak value of exciting voltage was 0.8 V [10]. Composite poly-Si&Si cantilevers HA_FM Etalon (ScanSens, Bremen, Germany) with a free resonance frequency of 80 kHz and a force constant of 3.5 N/m are used. The loading force is constant at ~120 nN.

*In situ synchrotron X-ray microdiffraction*. The measurement of the phase composition of the ferroelectric capacitor using in situ XRD using synchrotron radiation is performed at beamline P23 in the DESY synchrotron research facility in Germany. Notably, the X-ray beam is focused on the sample surface to a spot approximately 30 µm in diameter, which is less than the size of the top electrodes in our capacitor structure. The incidence angle of the X-ray beam was set to 15° in the sample plane. All measurements are performed with woken-up capacitors, which guarantees the end of possible structural transformations.

## 3. Results and Discussion

In the first step, we study the impact of the bottom electrode material (W vs. TiN) on inducing ferroelectricity in ultrathin HZO and select the most suitable one for further material engineering and investigation. At the same time, the top electrode is chosen to be the same throughout this work in order to separate its impact. TiN as the top electrode is chosen because of its excellent compatibility with modern microelectronics silicon technology [1,2,3]. We fabricate and compare PMA and PDA structures based on W and TiN bottom electrodes and HZO 10 and 5 nm in thickness. In this step, annealing is carried out at 500 and 600 °C.

Non-ultrathin 10 nm thick HZO films are taken for further comparison and analysis because it is optimal for inducing ferroelectricity as shown in many sources [1,2,3]. These 5.0 nm thick HZO films are chosen since this thickness is slightly larger than the expected critical thickness. Indeed, it was previously shown that the critical thickness at the silicon bottom electrode is 4.0 nm [17,18,19], below which the polarization drops sharply down to 1 µC/cm^2^ [17]. The last value corresponds to the density of the charging interface traps at the Si/HfO_2_ interface [15,16]. It should be noted that the use of silicon as a bottom electrode is not possible in some applications, such as back-end-of-line (BEoL) FRAM and FTJ. Regarding annealing temperature, we have previously established that the thinner the film, the larger the temperature budget required [17]. Indeed, while for the 10 nm thick HZO film, 400 °C is sufficient for inducing ferroelectricity; for thinner films, this temperature is no longer sufficient. Therefore, we use higher temperatures.

Comparing the switchable polarization values measured for the structures with W and TiN bottom electrodes after the wake-up procedure (Table 1 and Table 2), it can be noted that the W electrode better promotes ferroelectricity. This is true not only for the ultrathin (5.0 nm) HZO film but also for the 10 nm thick HZO film. This result is natural. As mentioned above, ferroelectricity in ultrathin HZO films is promoted by the use of substrates and electrode materials with a thermal coefficient of expansion (TCE) smaller than that for hafnium, and the smaller, the better. TCEs of the substrate and functional materials are equal to the following values: for Si substrate 2.6 × 10^−6^/°C [4], for HfO_2_ 8…10 × 10^−6^/°C [20], for W 4.5 × 10^−6^/°C [5], and for TiN 9.4 × 10^−6^/°C [5]. Therefore, during annealing, the Si substrate induces mechanical stress in HZO and facilitates the formation of the ferroelectric phase. The W bottom electrode leads to the same effect, whereas the TiN bottom electrode has almost the same TCE as HZO and thus suppresses the mechanical stress from the substrate.

The observed effect of the bottom electrode material is consistent with the results obtained earlier for HZO of other thicknesses [5,21]. Based on these results, the W bottom electrode is chosen for further study.

Looking ahead, it is worth noting that the W bottom electrode has not only advantages (promoting ferroelectricity and large remanent polarization) but also disadvantages. Namely, it leads to a relatively low endurance: the endurance of W/HZO (10 nm) structures is about 2 × 10^6^ switching cycles, while that of W/HZO (4.5 nm) is 4 × 10^5^ cycles. Low endurance is probably due to the relatively high roughness of W film. On the one hand, it is a quasi-3d bottom electrode in which the real area is larger than the nominal area and the measured polarization is larger. On the other hand, the developed surface leads to large variations of the electric field in the film (due to the effect of strengthening/weakening of the field by bulges and pits) and brings the electrical breakdown closer.

In the second step, we search for the minimum critical thickness of the HZO film grown on the metal W electrode. For this purpose, structures with film thicknesses of 4.0, 4.5, and 5.0 nm are fabricated under different crystallization conditions: PMA and PDA at 500, 600, and 700 °C. It is worth noting that we additionally increase the annealing temperature to increase the temperature budget.

Comparing the switchable polarization values measured for PMA and PDA structures after the wake-up procedure (Table 1 and Table 2), one could note that a 4.0 nm thick HZO film always has a very small switchable polarization, which resembles the charge density that can be injected into the interface traps [17]. In contrast, a slightly thicker film (4.5 nm) has significant polarization at high annealing temperatures (600 and 700 °C)—~23 µC/cm^2^ for PMA and 30 µC/cm^2^ for PDA. The switchable polarization for a 5.0 nm thick HZO film is almost equal to the polarization for a 4.5 nm thick film, so it can be concluded that 4.5 nm is the critical thickness for the HZO film grown on the W bottom electrode.

It is worth noting that the switchable polarization of the HZO (4.5 nm) crystallized via PDA is larger than that crystallized via PMA, though PMA is most commonly used. Before studying the origin of this result, in the third step, we analyzed the influence of annealing temperature on the functional properties. Figure 1a shows the dependences of switchable polarization and leakage current density on annealing temperature for the structure Si/W/HZO (4.5 nm)/TiN crystallized via PDA and PMA. For both types of annealing, the switchable polarization is constant at 600 and 700 °C and decreases at 500 °C, i.e., it stabilizes at ~600 °C. Meanwhile, for the 10 nm thick HZO film, the switchable polarization stabilizes at a lower temperature (500 °C). Thus, a higher temperature budget is needed to induce ferroelectricity in ultrathin films, but further increasing the temperature above the critical temperature has no effect on the remanent polarization.

On the other hand, an increase in temperature leads to an increase in leakage currents (Figure 1a). This parasitic effect is explained by the coarsening of grains accompanied by an increase in undesirable leakage currents flowing along grain boundaries. The second cause is defect generation, which mainly takes place at the metal electrode interfaces due to redox reactions. This explanation is supported by the results obtained, which show that the leakage currents increase with temperature more strongly in PMA structures. Indeed, unlike PDA structures, in PMA structures, annealing is carried out when the HZO layer is sandwiched between two metal electrodes and more oxygen vacancies are formed. To better understand the role of grains in leakage currents, the surface topography of HZO films (4.5 nm) subjected to PMA and PDA can be compared (Figure 1c). It is evident that the morphology and roughness of these films are similar. This implies similar grain size, which corresponds to similar leakage currents in these structures. To summarize, large leakage currents along grain boundaries and defects are undesirable because they increase the power consumption of FRAM and interfere with the resistive switching effect in FTJ.

Therefore, the optimal structure based on the ultrathin HZO film is Si/W/HZO (4.5 nm)/TiN crystallized via PDA. Note that the difference in the thickness of the studied films is very small. Indeed, 0.5 nm is about of lattice constant of hafnia. In order to control the thickness of the grown films, the most important samples from the set are examined by statistical analysis of cross-sectional high-resolution transmission electron microscopy (HRTEM) images. For example, Figure 2b shows the cross section of the Si/W/HZO (4.5 nm)/TiN structure crystallized via PDA. The thicknesses notated in this work were precisely obtained using HRTEM.

In the fourth step, to clarify the difference in the functional properties of PMA and PDA structures based on a 4.5 nm thick HZO film, they are studied in more detail. For more information, we also study similar structures based on a 10 nm thick HZO film. For 4.5 nm and 10 nm thick HZO films, we choose structures annealed at 600 and 500 °C, respectively, because these temperatures are optimal for these thicknesses.

Due to the small thickness of the ferroelectric layer, the current through the ferroelectric is quite high even in high-quality low-defect HZO films. This current contributes significantly to the measured *P–V* hysteresis, stretching it vertically and thus leading to incorrect values of the measured remanent polarization. To avoid this, we use a dynamic positive-up negative-down (PUND)-like technique that compensates for the contribution of leakage currents and obtains correct polarization values [10]. A typical voltage pulse set and current response are shown in Figure 2a. The input pulse set consists of two pairs of unipolar (positive and negative) triangular pulses. During the first pulse in the pair of identical unipolar pulses, the current *I*_sw_ is the sum of the current associated with polarization switching plus the leakage current and the capacitive displacement current, while the current during the second pulse *I*_non–sw_ has no polarization switching contribution and originates only from the leakage current and the capacitive displacement current. Thus, the polarization switching current can be found as the difference between the integral current signals generated by the first and second unipolar pulses. In turn, the *P–V* hysteresis is obtained by time integration of the polarization switching current. A PUND-like technique is used to measure the evolution of the remanent polarization during the wake-up process and the *P–V* hysteresis at different moments of the structure’s lifetime.

The evolution of the remanent polarization during the wake-up process differs for PMA and PDA samples (Figure 2c). The PDA structures based on 4.5 nm and 10 nm thick HZO films exhibit very similar behavior. Their remanent polarization is the same both at the end of and during the wake-up procedure. In contrast, the remanent polarization of the PMA structure based on HZO (4.5 nm) is much smaller than that based on HZO (10 nm). In addition, in the PDA structure based on HZO (4.5 nm), the remanent polarization stabilizes almost at the beginning of electrical cycling and grows very weakly thereafter, whereas in the analogous PMA structure, a noticeable growth is observed at the end of cycling and no polarization stabilization is observed at all.

To understand the difference in behavior, we analyze the evolution of the switching *I–V* and *P–V* curves during the wake-up. Very important information is contained in the shape of the very first two curves, i.e., the switching *I–V* and *P–V* curves corresponding to the first two write-rewrite cycles of the as-prepared structure. They reflect the state (mono- or polydomain state, fractions of domain) of the as-prepared structure as well as the magnitude and sign of the internal built-in field across the HZO layer. For example, splitting of the switching *I–V* curve and related pinched *P–V* curve observed for the PMA structure based on HZO (4.5 nm) indicates the existence of two opposite internal built-in fields (compare Figure 2b,d) [7], which are signatures of the polydomain structure in the as-prepared film. Specifically, the internal built-in fields originate from the spatial distribution of non-ferroelectric charges governed by the depolarization field (ferroelectric polarization vector) that has opposite directions in the domains with up- and downward polarization [11,12,13]. The polydomain structure that is peculiar to the PMA structure is caused by the charged oxygen vacancies formed at both bottom and top electrode interfaces during annealing because of redox reactions with the electrode materials. Depending on the mutual distribution of the oxygen vacancies at the two interfaces, HZO locally switches to up- or downward polarization during crystallization into the ferroelectric phase.

On the contrary, as-prepared structures that underwent PDA exhibit a non-split *I–V* curve and thus open *P–V* curve (Figure 2d). However, curves are shifted along the *V*-axis (the so-called imprint effect), indicating the presence of a unidirectional internal embedded field and a monodomain structure. This is caused by the formation of the charged oxygen vacancies at only one (bottom) interface.

After the wake-up procedure, the internal built-in field in the PDA structure disappears and the *I–V* and *P–V* curves become non-imprinted (symmetrical with respect to zero voltage) (Figure 2d). In contrast, for the PMA structure, the *I–V* curve remains to be split and the *P–V* curve remains to be pinched (Figure 2e). Therefore, two opposite internal built-in fields are retained in HZO (4.5 nm), though they decrease. As a result, even at the end of the wake-up procedure, some of the domains are pinned and the total remanent polarization remains to be smaller than that for 10 nm thick HZO (Figure 2b).

Next, one could analyze how the HZO thickness affects the built-in fields in the film. We estimate the magnitude of the built-in fields for each domain population using the sum of the average coercive voltages (Figure 2b), which corresponds to the shift of the switching *I–V* curves with respect to the zero field. For example, a shift of the *I–V* curves toward negative voltages on *V*_bi 1_ indicates the presence of a positive built-in field *V*_bi 1_ in a given domain population. From Figure 2d,e, one can see that an increase in the HZO thickness leads to a decrease in the built-in fields, which is reasonable because the electric field strength in a flat capacitor is inversely proportional to the distance between its plates. For PMA structures, the built-in fields decrease from 3.3 and −1.7 MV/cm for 4.5 nm to 2.3 and −0.5 MV/cm for 10 nm, and for PDA structures, the built-in field decreases from 2.0 MV/cm to 1.7 MV/cm.

It should be noted that in the structures where the *P–V* curve became open after the wake-up procedure, this curve is almost symmetric with respect to zero voltage (imprint not more than 0.2 MV/cm). This value probably corresponds to the potential difference between the W bottom and TiN top electrodes.

To better understand the influence of the internal built-in fields in PMA and PDA structures on their functional properties, we study the domain structure in capacitor devices and its transformation at different stages of capacitor lifetime by in situ piezoresponse force microscopy (PFM) [10]. Fundamentally, the PFM technique provides two informative quantities: the PFM amplitude *A*, associated with the absolute magnitude of the effective longitudinal piezoelectric coefficient (piezomodule) *d*_33_^*^, and the PFM phase φ, associated either with the orientation of the polarization vector (either up- or downward). Using *A* and φ, one could plot the maps of piezoresponse *A* sinφ, which visualize both the magnitude and orientation of the vertical component of the polarization.

First, we analyze the local ferroelectric properties of HZO at the initial stage of the memory cell life, namely, we compared the domain structures after the first switching pulses. For both PMA and PDA structures based on 10 nm thick HZO films, one could see the pinning of domains with both polarizations (Figure 3a,c), i.e., initially, some part of the polarization is not switched in both types of structures. The area (fraction) of pinned domains is much larger for the PMA structure, in which most of the remanent polarization is suppressed at the initial stage of the memory cell life.

Additional information can be obtained from the overlapping of PFM phase maps measured after switching with pulses of opposite signs (Figure 3a,c). For the PDA structure, the overlapping just confirms the presence of non-switching *P*↑ and *P*↓ domains (blue and orange-colored regions) and also shows that in most of the domains, the polarization vector switches to normal, i.e., along the applied field (purple-colored regions). The PMA structure additionally has domains of the fourth type that correspond to the local anomalous polarization switching (sienna-colored regions). Anomalous domains are regions where the polarization switches oppositely to the applied field and they are surrounded by the non (or back)—switching domains. Previously, such domains were also observed in structures with other electrodes during the wake-up [10].

The phenomenon of anomalous switching in hafnia originates from internal built-in fields caused by the agglomerates of the oxygen vacancies forming at the electrode interfaces during annealing due to redox reaction [10]. If the oxygen vacancy density is locally large at both interfaces, the halo of “anomalous” domains can be composite, i.e., it consists of non-switching domains with both up and downward polarization. This effect takes place when the areas of the oxygen vacancies agglomerated at both interfaces overlap in the substrate plane. During the wake-up, oxygen vacancies are gradually redistributed over the film bulk, their local density decreases and the internal built-in fields gradually disappear. As a result, the anomalous and pinned domains disappear, and the whole film begins to switch normally (Figure 2b,d).

It is remarkable that the phenomenon of anomalous switching is observed in the PMA structure and is not observed in the PDA structure. This indicates that regions of oxygen vacancy agglomeration at least at one of the interfaces are sparsely located. This effect is due to the absence of one (top) interface during annealing and the suppression of the redox reaction at it. Thus, the microscopic study confirms the critical role of internal built-in fields in the ferroelectric properties of PMA and PDA structures.

To get further insight into the functionality of PMA and PDA structures for memory applications, we analyze the magnitude of the longitudinal piezoelectric coefficient in cycled structures. After the wake-up procedure, both structures are in a monodomain state, i.e., domain pinning is completely vanishing, and polarization switches throughout the film (Figure 3b,d). The average piezomodule over the studied area of the PMA structure is 0.33 pm/V, which is in agreement with the previous synchrotron X-ray microdiffractometry results [22]. The average piezomodule of the PDA structure is 0.16 pm/V, i.e., half as much (Figure 3e). Meanwhile, the piezoelectric coefficient is proportionally related to the remanent polarization *P*_r_: *d*_33_ = 2ε_0_*kQ*_11_*P*_r_, where ε_0_ is the permittivity of the vacuum, *k* is the dielectric constant, and *Q*_11_ is the electrostrictive constant. Thus, a larger piezomodule corresponds to a larger vertical component of the polarization vector. In general, this is consistent with the remanent polarization 2*P*_r_ values obtained, which are 42 and 30 µC/cm^2^ for the PMA and PDA structures, respectively.

However, the difference between the values of the piezoelectric coefficient and remanent polarization is not the same. One possible reason could be the presence of a non-ferroelectric structural phase, such as a monoclinic phase. Although the PFM shows polarization switching throughout the film, individual grains or their small groups are not distinguishable due to mechanical coupling with neighboring ferroelectric grains [22]. To resolve this issue, we study the structural composition of both structures by synchrotron X-ray micro-diffraction.

The phase composition of ferroelectric structures Si/W/HZO (10 nm)/TiN that underwent PMA and PDA is measured using in situ XRD using synchrotron radiation. Notably, the X-ray beam is focused on the sample surface to a spot approximately 30 µm in diameter, which is less than the size of the top electrodes in the capacitor structures and ensures that the cycled HZO is under study. Measurements are performed with woken-up capacitors, which guarantee the end of possible structural transformations.

Comparison of full XRD spectra from PMA and PDA structures (Figure 4b) reveals that the main difference between PMA and PDA is in the 2θ range of 26–37° (besides the peak of orthorhombic phase at 60.69°). The decomposition of spectra in this range (Figure 4a) elucidates that the ratio of ferroelectric orthorhombic phase fraction to non-ferroelectric monoclinic phase fraction differs significantly. Specifically, the PMA sample contains more ferroelectric phase and less monoclinic phase than the PDA structure. This is in general agreement with their remanent polarizations and piezoelectric coefficients. However, the quantitative ratios of these values for the two types of samples differ less than the structural phase ratios. This may indicate that in the PDA structure, more grains of the ferroelectric phase are oriented more vertically. The spectra confirm this assumption. Indeed, in the second sample, the ratio of the intensity of peak o 002, corresponding to grains with vertical polar axis, to the intensity of peak o 111 is larger (0.243 vs. 0.188). Therefore, PDA processing possibly not only induces the formation of a large fraction of the monoclinic phase but also promotes the vertical texturing of ferroelectric grains.

In addition, it is useful to compare the lattice constants of the orthorhombic phase in PMA and PDA structures. Indeed, the ferroelectric orthorhombic and non-ferroelectric tetragonal phases of HfO_2_ and HZO are both slightly distorted cubic lattices. Therefore, even a minor change in the lattice parameter values may indicate a phase transition and, consequently, ferroelectric response change [23,24]. The calculated ratio of unit cell parameters *c*/*a* (*c* is polar axis) turns out to be also slightly larger in the case of PDA (0.988 vs. 0.969), i.e., a non-monoclinic fraction of the PDA film is closer to the ferroelectric orthorhombic phase. Thus, it has been experimentally established that PDA induces the formation of a ferroelectric phase that gives a larger contribution to the measured remanent polarization. The reason for this is currently unknown and further studies are required.

It should be noted that the monoclinic phase has a narrower band gap [17]. Therefore, the grains of the monoclinic phase are sites of leakage current flowing. This parasitic current cannot be modulated by polarization switching in the neighboring grains of the ferroelectric phase. Therefore, HZO films with a large fraction of the monoclinic phase are poorly suited for the development of FTJ devices. However, ultra-thin HZO films, even if they have similar structural properties, are promising for developing low-power and high-density FRAM devices. In addition, our findings may be useful for further development of ultrathin ferroelectric films for FTJ and memristor devices.

## 4. Conclusions

We report an approach to fabricate the wake-up free ferroelectric memory cells based on ultrathin (4.5 nm) HZO film and reveal its physical basis. The approach relies on the suppression of internal built-in fields, which can be very large in ultrathin films and lead to strong domain pinning. This is realized via post-deposition annealing, i.e., without a top electrode that leads to the generation of charged defects at the top interface. This assumption is confirmed by investigating the domain structure transformation by in situ PFM.

By comparison of the functional properties with piezoelectric properties of the woken-up capacitor resolved with in situ PFM and structural phase composition acquired by in situ synchrotron micro-XRD, we clarify the role of post-deposition and post-metallization annealing in the ferroelectric properties of HZO. We show that the parasitic contribution of the internal electric fields increases with the thickness decrease. In PMA structures, the contribution of the internal electric fields is always larger. However, in 10 nm thick films, it does not reach the threshold value when its parasitic contribution suppresses the switchable polarization, but in ultrathin films, it reaches this threshold value. The use of PDA cancels this challenge.

The proposed approach could be useful not only for the design of conventional high-density FRAM and FTJ memory but also for the development of novel devices based on tunnel-transparent ferroelectrics, which require functional materials that suppress ferroelectricity in hafnium oxide (e.g., graphene and other two-dimensional materials [25,26,27]).

## Figures and Tables

**Figure 1 nanomaterials-13-02825-f001:**
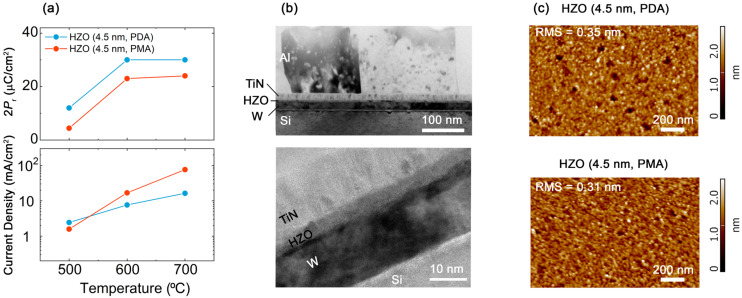
(**a**) Functional properties of Si/W/HZO (4.5 nm)/TiN crystallized via PDA and PMA. Leakage currents are measured at 4 MV/cm. Both 2*P*_r_ and leakage current are measured after the wake-up procedure. (**b**) Cross-sectional HRTEM images of Si/W/HZO (4.5 nm)/TiN crystallized via PDA. (**c**) Topography of Si/W/HZO (4.5 nm) surface measured by atomic force microscopy after TiN electrode patterning. The pits are due to the inhomogeneous etching of HZO, which had time to start immediately after the completion of TiN etching.

**Figure 2 nanomaterials-13-02825-f002:**
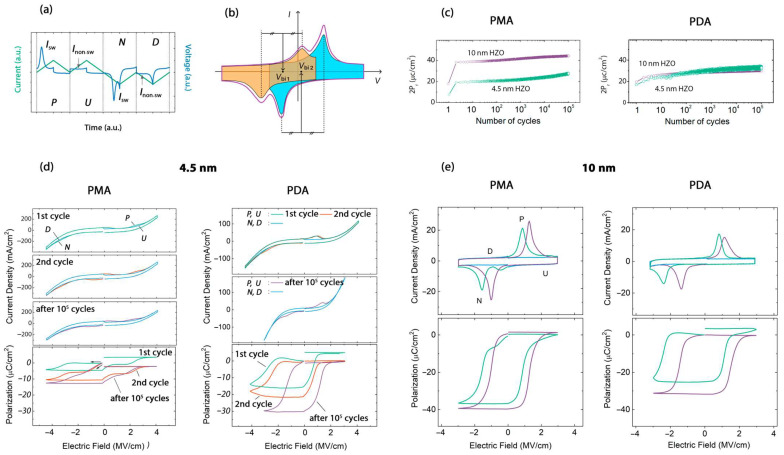
Functional properties of the PMA and PDA structures based on 4.5 and 10 nm thick HZO. (**a**) Voltage pulse sequences and current response in PUND-like technique. (**b**,**c**) Evolution of the remanent polarization during the wake-up procedure. (**d**) Switching *I–V* and *P–V* curves of the samples based on 4.5 nm thick film. (**e**) Switching *I–V* and *P–V* curves of the samples based on 10 nm thick film.

**Figure 3 nanomaterials-13-02825-f003:**
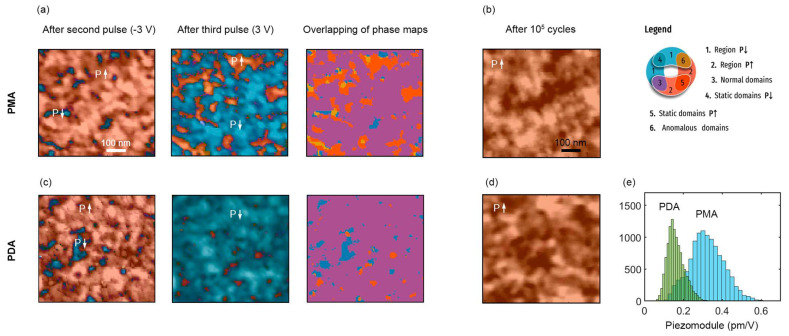
Domain structures of PMA and PDA structures at different stages of their lifetime. (**a**,**c**) Piezoresponse maps of as-prepared PMA and PDA structures after first positive and negative switching pulses and overlapping of PFM phase maps. (**b**,**d**) Piezoresponse maps of cycled PMA and PDA structures after the voltage pulse 3 V/10 µs. (**e**) Distributions of piezomodule over maps (**b**,**d**).

**Figure 4 nanomaterials-13-02825-f004:**
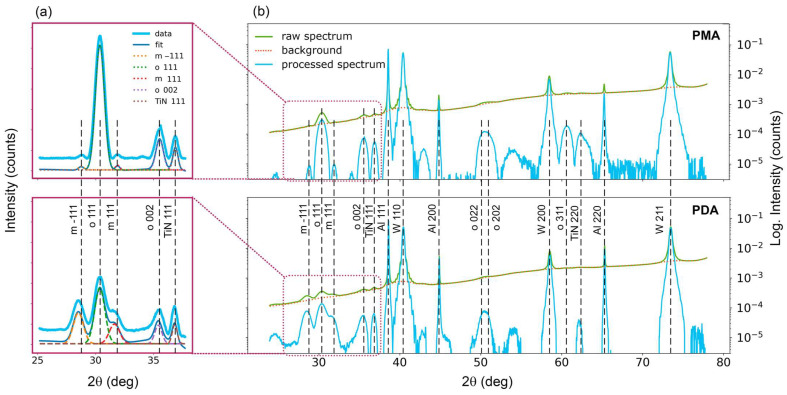
Comparison of the structural phase composition in PMA and PDA structures by synchrotron X-ray micro-diffraction (micro-XRD). (**a**) XRD spectra in the 2θ range of 26–37 degrees (light blue lines are vertically shifted experimental spectra, dotted lines are fitting peaks, and the blue line is the sum of fitting peaks). (**b**) Full XRD spectra in semi-logarithmic scale.

**Table 1 nanomaterials-13-02825-t001:** Remanent polarization of the PMA structures measured after the wake-up procedure.

Structure		W/HZO/TiN				TiN/HZO/TiN
	Thickness nm	4.0	4.5	5.0	10	5.0
Ann Temp. °C						
500		1	4	6	40	1
600		4	23	25	42	7
700		too leaky	24	27	-	-

**Table 2 nanomaterials-13-02825-t002:** Remanent polarization of the PDA structures measured after the wake-up procedure.

Structure		W/HZO/TiN				TiN/HZO/TiN
	Thickness nm	4.0	4.5	5.0	10	5.0
Ann. Temp. °C						
500		1	12	14	29	1
600		too leaky	30	30	29	20
700		too leaky	30	30	-	-

## Data Availability

Not applicable.

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
