# Peer review of "Wake-Up Free Ultrathin Ferroelectric Hf0.5Zr0.5O2 Films"

_nanomaterials, 2023, doi:10.3390/nano13212825_

Round 1

Reviewer 1 Report

The paper is of high interest, dealing with the hot topic of ferroelectric HZO with important results related to obtaining wake-up free ultrathin ferroelectric HZO films (4.5 nm thickness).

Bellow are my comments:

-        PFM and XRD results on ultrathin ferroelectric HZO (4.5 nm) are missing. PFM and XRD conclusions are drawn only for 10 nm HZO.

-        PMA vs PDA in structures with 10 nm HZO: PMA is the best as it gives the highest remanent polarization. For very thin films (4.5 and 6 nm) it is the opposite. This is different compared with literature that reports PMA as being better than PDA for obtaining improved ferroelectric properties.

-        In 2Pr function of number of cycles: what happens after 106 cycles?

-        It is reported that large in-plane tensile stress is induced in HZO thin and ultrathin films by W bottom electrode that has low thermal expansion coefficient [Appl. Phys. Lett. 117, 242901 (2020); doi: 10.1063/5.0029516] leading to formation of ferroelectric o-phase.

-        Due to difference in work functions of bottom and top electrodes, built-in potential is generated. Please, discuss

-        This may indicate that in the PDA sample more grains of ferroelectric phase are oriented more vertically”; “PDA processing promotes the vertical texturing of ferroelectric grains” – it is known that “in a 5 nm-thick HfO2 film, only ten unit cells of the HfO2 crystal are stacked vertically”. Please detail the mechanism for promoting the vertical texturing of ferroelectric grains

-        Evidence the role of Al layer in the structure by endurance

-        Please check citation of a-e in Figure2a-e and in text

Author Response

The paper is of high interest, dealing with the hot topic of ferroelectric HZO with important results related to obtaining wake-up free ultrathin ferroelectric HZO films (4.5 nm thickness).

Below are my comments:

We appreciate the comprehensive review and valuable comments from the Reviewer.

- PFM and XRD results on ultrathin ferroelectric HZO (4.5 nm) are missing. PFM and XRD conclusions are drawn only for 10 nm HZO.

We agree that it would be useful to have data for ultrathin ferroelectric HZO (4.5 nm) films as well. However, due to the very small thickness, such data would not be reliable, in PFM due to high leakage currents, and in XRD due to a significantly reduced XRD pattern intensity.

- PMA vs PDA in structures with 10 nm HZO: PMA is the best as it gives the highest remanent polarization. For very thin films (4.5 and 6 nm) it is the opposite. This is different compared with literature that reports PMA as being better than PDA for obtaining improved ferroelectric properties.

We agree that the performance of ultrathin films is rather unexpected. The ferroelectric properties of hafnium oxide depend on many factors (mechanical stresses, physical and chemical properties of the electrode interfaces, defect concentration, etc.) An increase or decrease in the role of different factors is to be expected with thickness variation. In this work, we show that the parasitic contribution of the internal electric fields increases with the thickness decrease. In PMA structures the contribution of the internal electric fields is always larger. However, in 10 nm thick films it does not reach the threshold value when its parasitic contribution suppresses polarization, but in ultrathin films it reaches this threshold value.

We have added a comment on this issue to the section “4. Conclusions”, p. 10-11: “We show that the parasitic contribution of the internal electric fields increases with the thickness decrease. In PMA structures, the contribution of the internal electric fields is always larger. However, in 10 nm thick films it does not reach the threshold value when its parasitic contribution suppresses the switchable polarization, but in ultrathin films it reaches this threshold value. The use of PDA cancels this challenge.”

- In 2Pr function of number of cycles: what happens after 106 cycles?

We have added a detailed comment on this issue (p. 4-5): “Looking ahead, it is worth noting that the W bottom electrode has not only ad-vantages (promoting ferroelectricity and large remanent polarization), but disadvantages too. Namely, it leads to a relatively low resource: the resource of W/HZO(10 nm) structures is about 2 x 106 switching cycles, while that of W/HZO(4.5 nm) is 4 x 105 cycles. This is probably due to the relatively high roughness of W film. On the one hand, it is a quasi-3d bottom electrode in which the real area is larger than the nominal area and the measured polarization is larger [14]. On the other hand, the developed surface leads to large variations of the electric field in the film (due to the effect of strengthening/weakening of the field by bulges and pits) and brings the electrical breakdown closer.”

- It is reported that large in-plane tensile stress is induced in HZO thin and ultrathin films by W bottom electrode that has low thermal expansion coefficient [Appl. Phys. Lett. 117, 242901 (2020); doi: 10.1063/5.0029516] leading to formation of ferroelectric o-phase.

Our conclusions are in agreement with this paper (ref. 5). Please find our comment on this issue on p. 4: “As mentioned above, ferroelectricity in ultrathin HZO films is promoted by the use of substrates and electrode materials with a thermal coefficient of expansion (TCE) smaller than that for hafnium, and the smaller the better. TCEs of substrate and functional materials are equal to the following values: for Si substrate 2.6 x 10-6/°C [4], for HfO2 8…10 x 10-6/°C [18], for W 4.5 x 10-6/°C [5], for TiN 9.4 x 10-6/°C [5]. Therefore, during annealing, Si substrate induces the mechanical stress in HZO and facilitates the formation of the ferroelectric phase. The W bottom electrode leads to the same effect, whereas the TiN bottom electrode has almost the same TCE as HZO and thus suppresses the mechanical stress from the substrate.

The observed effect of the bottom electrode material is consistent with the results ob-tained earlier for HZO of other thicknesses [5, 19].”

- Due to difference in work functions of bottom and top electrodes, built-in potential is generated. Please, discuss

We have added a comment on this issue, p. 8: “It should be noted that in the structures where the P-V curve became open after the wake-up procedure, this curve is almost symmetric with respect to zero voltage (imprint not more than 0.2 MV/cm). This value probably corresponds to the potential difference between the W bottom and TiN top electrodes.”

- “This may indicate that in the PDA sample more grains of ferroelectric phase are oriented more vertically”; “PDA processing promotes the vertical texturing of ferroelectric grains” – it is known that “in a 5 nm-thick HfO2 film, only ten unit cells of the HfO2 crystal are stacked vertically”. Please detail the mechanism for promoting the vertical texturing of ferroelectric grains

We are grateful for this comment. Indeed, the origin of vertical film texturing is an important and complex issue. It requires a bulk of additional studies and at present, we do not have an answer to this question. In fact, we simply describe what we see in the XRD spectra. In the revised manuscript, we have described this point in a non-affirmative way (p. 10).

- Evidence the role of Al layer in the structure by endurance.

We have re-write an appropriate text in order to make it more clear, p. 3: “Second, an additional Al layer 150 nm in thickness is deposited on the top of the TiN electrode using electron beam evaporation. Al layer minimizes RC time constant and thus allows to decrease the duration of switching pulses, which, in turn, improves the endurance of ferroelectric devices.”

- Please check citation of a-e in Figure2a-e and in text

Thank you very much for this comment. We have fixed appropriate typos.

Reviewer 2 Report

This is an excellent paper that addresses the wake-up effect in HZO dielectric thin film with thickness less than 5nm. The authors systematically studied the wave-up effect by selecting different electrodes, thin film thickness, and annealing temperature. The mechanical was disclosed. The paper is very interesting. I have only minor comment. Please provide the Atomic fore images of the HZO thin films with different conditions above. 

Author Response

This is an excellent paper that addresses the wake-up effect in HZO dielectric thin film with thickness less than 5nm. The authors systematically studied the wave-up effect by selecting different electrodes, thin film thickness, and annealing temperature. The mechanical was disclosed. The paper is very interesting. I have only minor comment.

- Please provide the Atomic fore images of the HZO thin films with different conditions above. 

We are grateful for the encouraging evaluation of our work. We have added AFM data and comment them (Figure 1c and an appropriate text, p. 6): “To better understand the role of grains in leakage currents, the surface topography of HZO films (4.5 nm) subjected to PMA and PDA can be compared (Figure 1c). It is evident that the morphology and roughness of these films are similar. This implies similar grain size, which corresponds to close leakage currents in these structures.”

Reviewer 3 Report

This paper talks about how the new generation of high-density ferroelectric memory relies on ultrathin ferroelectric films, with doped HfO2 films being the top contenders due to their compatibility with silicon technology and superior ferroelectric properties. By engineering internal built-in fields, stable ferroelectricity is achieved in a 4.5 nm thick Hf0.5Zr0.5O2 film.

This paper still has minor issues as listed below:

1. The head of the first column of Table 1 has a formatting error, please correct it. 

2. Does lattice parameter and crystallographic structure play an effect on the magnetic properties? 

Some sentences need to be modified. 

Author Response

This paper talks about how the new generation of high-density ferroelectric memory relies on ultrathin ferroelectric films, with doped HfO2 films being the top contenders due to their compatibility with silicon technology and superior ferroelectric properties. By engineering internal built-in fields, stable ferroelectricity is achieved in a 4.5 nm thick Hf0.5Zr0.5O2 film.

This paper still has minor issues as listed below:

  1. The head of the first column of Table 1 has a formatting error, please correct it. 

Thank you for this comment. Done.

  1. Does lattice parameter and crystallographic structure play an effect on the magnetic (ferroelectric) properties? 

Yes, both o- and t- phases in HfO2 and HZO are a slightly distorted cubic lattice. Therefore, even a minor change in the lattice parameter values may indicate a phase transition and, consequently, ferroelectric response change [C. Richter et al., https://doi.org/10.1002/aelm.201700131, M. G. Kozodaev et al., https://doi.org/10.1063/1.5050700]. The grain orientation also significantly affects the measured remanent polarization [M. H. Park et al., https://doi.org/10.1063/1.4866008].

In the revised manuscript, we have added a discussion on this issue (p. 10): “In addition, it is useful to compare the lattice constants of orthorhombic phase in PMA and PDA structures. Indeed, the ferroelectric orthorhombic and non-ferroelectric tetragonal phases of HfO2 and HZO are both a slightly distorted cubic lattice. Therefore, even a minor change in the lattice parameter values may indicate a phase transition and, consequently, ferroelectric response change [21]. The calculated ratio of unit cell parameters c/a (c is polar axis) turns out to be also slightly larger in the case of PDA (0.988 vs. 0.969), i.e., non-monoclinic fraction of the PDA film is closer to the ferroelectric orthorhombic phase. Thus, it has been experimentally established that PDA induces the formation of a ferroelectric phase that gives a larger contribution to the measured residual polarization. The reason for this is currently unknown and further studies are required.”

Round 2

Reviewer 1 Report

Accept in present form